# Nitrite Concentration in the Striated Muscles Is Reversely Related to Myoglobin and Mitochondrial Proteins Content in Rats

**DOI:** 10.3390/ijms23052686

**Published:** 2022-02-28

**Authors:** Joanna Majerczak, Agnieszka Kij, Hanna Drzymala-Celichowska, Kamil Kus, Janusz Karasinski, Zenon Nieckarz, Marcin Grandys, Jan Celichowski, Zbigniew Szkutnik, Ulrike B. Hendgen-Cotta, Jerzy A. Zoladz

**Affiliations:** 1Chair of Exercise Physiology and Muscle Bioenergetics, Faculty of Health Sciences, Jagiellonian University Medical College, 31-066 Krakow, Poland; marcin.grandys@uj.edu.pl (M.G.); j.zoladz@uj.edu.pl (J.A.Z.); 2Jagiellonian Centre for Experimental Therapeutics (JCET), Jagiellonian University, 30-348 Krakow, Poland; agnieszka.kij@uj.edu.pl (A.K.); kamil.kus@jcet.eu (K.K.); 3Department of Neurobiology, Faculty of Health Sciences, Poznan University of Physical Education, 61-871 Poznan, Poland; drzymala@awf.poznan.pl (H.D.-C.); celichowski@awf.poznan.pl (J.C.); 4Department of Physiology and Biochemistry, Faculty of Health Sciences, Poznan University of Physical Education, 61-871 Poznan, Poland; 5Department of Cell Biology and Imaging, Institute of Zoology and Biomedical Research, Jagiellonian University, 30-387 Krakow, Poland; janusz.karasinski@uj.edu.pl; 6Experimental Computer Physics Department, Marian Smoluchowski Institute of Physics, Jagiellonian University, 30-348 Krakow, Poland; zenon.nieckarz@uj.edu.pl; 7Faculty of Applied Mathematics, AGH-University of Science and Technology, 30-059 Krakow, Poland; szkutnik@agh.edu.pl; 8Department of Cardiology and Vascular Medicine, West German Heart and Vascular Center, University Hospital Essen, Medical Faculty, University of Duisburg-Essen, 45147 Essen, Germany; ulrike.hendgen-cotta@uk-essen.de

**Keywords:** nitrate, nitric oxide, heart, locomotory muscles, oxidative stress

## Abstract

Skeletal muscles are an important reservoir of nitric oxide (NO^•^) stored in the form of nitrite [NO_2_^−^] and nitrate [NO_3_^−^] (NO_x_). Nitrite, which can be reduced to NO^•^ under hypoxic and acidotic conditions, is considered a physiologically relevant, direct source of bioactive NO^•^. The aim of the present study was to determine the basal levels of NO_x_ in striated muscles (including rat heart and locomotory muscles) with varied contents of tissue nitrite reductases, such as myoglobin and mitochondrial electron transport chain proteins (ETC-proteins). Muscle NO_x_ was determined using a high-performance liquid chromatography-based method. Muscle proteins were evaluated using western-immunoblotting. We found that oxidative muscles with a higher content of ETC-proteins and myoglobin (such as the heart and slow-twitch locomotory muscles) have lower [NO_2_^−^] compared to fast-twitch muscles with a lower content of those proteins. The muscle type had no observed effect on the [NO_3_^−^]. Our results demonstrated that fast-twitch muscles possess greater potential to generate NO^•^ via nitrite reduction than slow-twitch muscles and the heart. This property might be of special importance for fast skeletal muscles during strenuous exercise and/or hypoxia since it might support muscle blood flow via additional NO^•^ provision (acidic/hypoxic vasodilation) and delay muscle fatigue.

## 1. Introduction

Nitric oxide (NO^•^) is a small, free radical and highly diffusible signalling molecule that at physiological concentrations (pM/nM) regulates vascular tone and exerts positive effects on neurotransmission, cell apoptosis, gene expression and immune response [1]. Conversely, NO^•^ at higher concentrations (>1 µM), e.g., during inflammatory states, can lead to oxidative, nitrative and nitrosative stress [2]. One of the most prominent effects of NO^•^ under physiological conditions (mediated by soluble guanylyl cyclase (sGC)) is related to its vasodilatory function, attenuation of vascular fibrosis, arterial stiffness and platelet aggregation [1,2]. In turn, a decrease in NO^•^ bioavailability in the circulatory system, caused by endothelial dysfunction, even under physiological conditions such as aging [3], leads to hyperinflammatory, hypercoagulative, and vasoconstrictive states in vessels, all of which contribute to arterial stiffness and increased risk of cardiovascular disease (CVD) [4]. It is worth mentioning that NO^•^ reversibly inhibits cytochrome-*c* oxidase (competitive with oxygen) under physiological conditions, leading to increased reduction of electron transport chain (ETC) and formation of reactive oxygen species (ROS) [5,6]. Therefore, NO^•^ (competitive with oxygen) might control oxidative phosphorylation (OXPHOS) efficiency and ROS generation in mitochondria [6].

The production of NO^•^ is largely covered by the classical pathway from L-arginine and oxygen in the presence of NO^•^ synthases (NOS-dependent pathway). NOS are present in three different isoforms: neuronal (NOS1), inducible (NOS2) and endothelial (NOS3) [7]. A majority of NO^•^ produced in the classical pathway becomes oxidised to nitrate (NO_3_^−^) (i.e., reaction with oxygenated haem proteins) and nitrite (NO_2_^−^) (i.e., reaction with ceruloplasmin in plasma and cytochrome-*c* oxidase in tissues). Interestingly, NO_2_^−^ and NO_3_^−^, regarded as useless end-products of NO^•^ oxidation, constitute substrates for NO^•^ generation through the reduction reaction in the non-classical, NOS-independent pathway (nitrate–nitrite–NO^•^ pathway) [8]. It needs to be underlined that tissue NO_3_^−^ concentration is several times higher than that of NO_2_^−^ [9,10,11], however NO_2_^−^ as a direct source of NO^•^, especially under acidic/reducing conditions [8,12,13] seems to be a more important player in NO^•^ homeostasis than NO_3_^−^. As indicated, NO_2_^−^ reduction represents a physiological mechanism by which NO^•^ production can be sustained under challenging conditions, especially hypoxia when NO^•^ generation by NOS is compromised [8,13]. Furthermore, the importance of tissue NO_2_^−^ content is related to the fact that, even under physiological conditions, NO_2_^−^ mediates cellular signalling through mechanisms, which are independent from those mediated by NO^•^ [14]. Interestingly, it was nitrite, not nitrate supplementation, that attenuated oxidative stress, reduced inflammation and ameliorated endothelial dysfunction in hypertensive mice [15].

It has been demonstrated that mammalian tissues (including blood and muscles) possess inherent nitrate/nitrite reductase activity [16]. The reduction of NO_2_^−^ to bioactive NO^•^ occurs through acidification or by the involvement of several NO_2_^−^ reductases including haem globins (haemoglobin, myoglobin, neuroglobin, cytoglobin), molybdenum-containing enzymes (xanthine oxidoreductase, aldehyde oxidase) [13] and ETC (Q-cycle and cytochrome-*c* oxidase) in mitochondria [17].

The importance of NO^•^ and NO_x_ (nitrite and nitrate) for striated muscle function remains to be elucidated. It has been found that an increase in NO^•^ bioavailability (nitrate/nitrite supplementation) potentiates the contractile properties of skeletal muscle [18] and the heart [19], even in patients with heart failure [20]. In addition, it has been reported that nitrate supplementation reduces oxygen cost at submaximal exercise [21] and improves mitochondrial efficiency in locomotory muscles [22].

Skeletal muscles have been found to be an endogenous reservoir of NO^•^ in the form of NO_3_^−^ and NO_2_^−^ [10]. Since skeletal muscles account for about 30–40% of body mass [23], their NO_3_^−^ and NO_2_^−^ pool might exert a substantial impact on NO^•^ bioavailability, not only for the cardiovascular system (blood flow regulation) but also for skeletal muscles in situ (contractility and energy metabolism) [18]. Due to the different physiological characteristics of various muscle fibre types, it is important to know whether nitrate/nitrite reservoirs in individual muscles–and their potential to generate the NO^•^ via the NO_2_^−^ reduction reaction–are similar in slow versus fast mammalian skeletal muscles. It has been postulated that an enhancement of NO^•^ bioavailability through nitrate supplementation might be especially effective in fast-twitch muscle fibres [24]. Specifically, it has been demonstrated that nitrate supplementation preferentially increases blood flow in exercising muscles composed predominantly of fast-twitch muscle fibres [25]. In addition, such supplementation enhances force production in fast-twitch muscle fibres, but not in slow-twitch muscle fibres [18].

The present study hypothesised that the ability to generate NO^•^ via the NO_2_^−^ reduction reaction (assessed by NO_2_^−^ concentration in the muscle as the substrate available) might differ between fast and slow muscles, as suggested previously [24]. We have conjectured that fast-twitch, glycolytic muscles, which are predominantly recruited at high power outputs [26], developing profound metabolic perturbations including deep acidosis and inorganic phosphate accumulation—key factors causing muscle fatigue [27,28,29], may possess a greater capacity than slow-twitch oxidative muscles to generate NO^•^ via NO_2_^−^ reduction as a fatigue compensation mechanism. The significance of NO^•^ production through the nitrite reduction might rely on the blood flow regulation during exercise to the fast muscle fibres [25] on the one hand, and on maintaining power generating capabilities of the fast muscles during high-intensity exercise on the other hand [18]. Therefore, to test the hypothesis of the importance of the nitrite reduction in varied muscle types, in the present study we evaluated NO_2_^−^ and NO_3_^−^ in rat striated muscle, i.e., in skeletal muscles with varied muscle fibre type compositions, namely the soleus (Sol), medial gastrocnemius (MG), the tibialis anterior (TA) and in the heart ventricles. The MG was divided by excision into a red part (MGS, slow medial gastrocnemius) and a white part (MGF, fast medial gastrocnemius). Moreover, we aimed to relate NO_2_^−^ concentration in striated muscles (as a stable, direct source of NO^•^) to the muscle content of myoglobin and ETC proteins, which are involved not only in oxygen turnover (oxygen buffering and oxygen usage in OXPHOS) but also in NO^•^ homeostasis [13,17].

## 2. Results

### 2.1. Nitrite and Nitrate Concentrations in the Heart and in the Skeletal Muscles with Varied Muscle Fibre Type Composition

Nitrite concentration (relative to total protein concentration, [NO_2_^−^]) was found to be significantly lower in the heart compared to the locomotory muscles, i.e., slow-twitch soleus (Sol, *p* = 0.02) and fast-twitch muscles such as MGS (*p* = 0.02), TA (*p* = 0.001) and MGF (*p* = 0.001) (Figure 1A). Moreover, in the locomotory muscle group, [NO_2_^−^] was significantly higher in the fast-twitch TA (~2.7-fold, *p* = 0.001) than in the slow-twitch Sol. In addition, [NO_2_^−^] was significantly higher in the fast-twitch MGF (~1.3-fold, *p* = 0.003) than in the Sol. No significant difference (*p* > 0.05) in [NO_2_^−^] was observed between MGF and MGS, MGS and TA, or MGF and TA (Figure 1A).

Muscle type was found to have a significant impact on the nitrate level (ANOVA, *p* = 0.04, Figure 1B). However, post-hoc analysis revealed only a tendency for higher [NO_3_^−^] in the MGS compared to the heart (*p* = 0.06) and Sol (*p* = 0.06) (Figure 1B).

### 2.2. Myoglobin and ETC Proteins Content in the Heart and in the Skeletal Muscles with Varied Muscle Fibre Type Composition

The level of the oxygen buffering protein myoglobin in the heart was similar to that in Sol (*p* > 0.05), however, it was significantly higher than in MGS (~1.8-fold, *p* = 0.007) and TA (~4.5-fold, *p* = 0.0001) (Figure 1C). Moreover, a clear tendency towards a higher myoglobin content in the heart compared to MGF was observed (*p* = 0.053). Furthermore, myoglobin content in the slow-twitch Sol was significantly higher than in fast oxidative MGS (~1.5-fold, *p* = 0.02) and fast glycolytic TA (~3.8-fold, *p* = 0.0004). The difference between the Sol and MGF was not statistically significant (~2-fold, *p* = 0.14). Additionally, no significant differences (*p* > 0.05) in myoglobin content between MGS and TA, MGS and MGF, or MGF and TA were found (Figure 1C).

For this study, the mitochondrial ETC proteins subunits were presented as the sum of the subunit of complex II, complex III, complex IV and complex V (Appendix A). We found that the sum of mitochondrial ETC proteins content (Figure 1D) was significantly higher in the heart than in skeletal muscles: Sol (~3.4-fold, *p* = 0.001), MGS (~6-fold, *p* = 0.001), TA (~9-fold, *p* = 0.001) and MGF (~11-fold, *p* = 0.001). Furthermore, in the skeletal muscle, the ETC protein content in the slow-twitch Sol was significantly higher than in MGS (~1.8-fold, *p* = 0.001), TA (~2.7-fold, *p* = 0.001) and MGF (~3.4-fold, *p* = 0.001). In turn, the ETC protein content was significantly higher in the MGS than in TA (*p* = 0.005) and MGF (*p* = 0.001). Moreover, ETC protein content in TA was significantly higher than in MGF (*p* = 0.007).

Additionally, based on the report presented by Larsen et al. [30] showing that the protein content of complexes II and V seemed to be relevant representatives of the mitochondrial content, we present these proteins (subunits of complex II and V) in the Appendix A.

### 2.3. NO^•^ Synthase and Arginase Activities in the Heart and in the Skeletal Muscles with Varied Muscle Fibre Type Composition

The conventional pathway of NO^•^ synthesis involves arginine, oxygen and NOS activity. Figure 2 demonstrates the ratio of citrulline-to-arginine in muscle, which reflects global NOS activity (Figure 2A), and the ratio of ornithine-to-arginine, which reflects arginase activity (Figure 2B).

NOS activity (citrulline-to-arginine ratio) in the heart was not found to be significantly different to the slow-twitch Sol (*p* > 0.05). However, it was significantly lower compared to fast locomotory muscles: MGS (*p* = 0.0001), TA (*p* = 0.07) and MGF (*p* = 0.0001). Moreover, in the group of locomotory muscles, NOS activity in the Sol was significantly lower than in the fast-twitch MGS (*p* = 0.0002) or MGF (*p* = 0.002). No significant difference in the NOS activity between Sol and TA was found (*p* > 0.05). In addition, a significant difference between TA and MGS (*p* = 0.002) and MGF (*p* = 0.02) was observed.

The ornithine-to-arginine ratio (Figure 2B), reflecting arginase activity, was significantly higher in the fast-twitch TA compared to the heart muscle (*p* = 0.04) and MGF (*p* = 0.001). No significant difference (*p* > 0.05) in the arginase activity between heart and other locomotory muscles was found.

### 2.4. Oxidative Stress and Antioxidant Capacity Markers in the Heart and in the Skeletal Muscles with Varied Muscle Fibre Type Composition

At basal conditions, the level of oxidative stress reflected by the content of the oxidised form of glutathione ([GSSG]) was significantly higher in the heart and in oxidative slow-twitch Sol compared to fast-twitch muscles such as MGS, TA or MGF (Figure 3A). Specifically, [GSSG] concentration in the heart was significantly higher than in the Sol (*p* = 0.001), MGS (*p* = 0.005), TA (*p* = 0.001) and MGF (*p* = 0.001). Furthermore, [GSSG] in the Sol was significantly higher than in MGS (*p* = 0.001), TA (*p* = 0.001) and MGF (*p* = 0.001). In addition, a higher content of [GSSG] in MGS compared to TA (*p* = 0.005) and MGF (*p* = 0.001) was found.

The antioxidant capacity of striated muscles was described by the content of the reduced form of glutathione ([GSH]), [GSH-to-GSSG] ratio and superoxide dismutase 2 (SOD2) expression. Specifically, the content of reduced glutathione ([GSH]) was significantly lower (*p* < 0.01) in the heart compared to the Sol (*p* = 0.001), MGS (*p* = 0.005), MGF (*p* = 0.001) and TA (*p* = 0.001) (Figure 3B). In the locomotory muscle group, [GSH] was significantly higher in the slow-twitch Sol than in the fast locomotory muscles: MGS (*p* = 0.001), MGF (*p* = 0.001) and TA (*p* = 0.001). In the fast muscle group, [GSH] was lower in MGS compared to TA (*p* = 0.005) and MGF (*p* = 0.001). No significant difference (*p* > 0.05) in the [GSH] was noticed between TA and MGF.

The antioxidant capacity of the tissue, which is reflected by the [GSH-to-GSSG] ratio, was lower (*p* < 0.01) in the heart compared to locomotory muscles: Sol (*p* = 0.001), MGS (*p* = 0.001), TA (*p* = 0.001) and MGF (*p* = 0.001) (Figure 3C). In the locomotory muscle group, the [GSH-to-GSSG] ratio in the Sol was significantly higher than in MGS (*p* = 0.001) and MGF (*p* = 0.02). No significant difference (*p* > 0.05) in the [GSH-to-GSSG] ratio was noticed between the Sol and TA. In the fast locomotory muscle group, a significantly lower [GSH-to-GSSG] ratio was found in the MGS compared to MGF (*p* = 0.001) and TA (*p* = 0.001).

In addition, we evaluated the expression of the mitochondrial isoform of SOD2 in the striated muscles (Figure 3D). Specifically, SOD2 concentration in the heart was found to be significantly higher than in MGS (~4.0-fold, *p* = 0.0001), TA (~4.5-fold, *p* = 0.0001) and MGF (~14-fold, *p* = 0.0001). No significant difference in SOD2 content was found between the heart and Sol muscles (*p* > 0.05). In the locomotory muscle group, significantly higher SOD2 content was found in the Sol compared to MGS (~2.7-fold, *p* = 0.0001), TA (~3.0-fold, *p* = 0.0001) and MGF (~9.7-fold, *p* = 0.0001). No significant difference was observed between MGS and TA, whereas SOD2 content was significantly higher in the MGS compared to MGF (*p* = 0.0001) and TA (*p* = 0.0001, Figure 3D).

These results demonstrate that [GSSG] and the content of SOD2 in the striated muscles under basal conditions follow a similar pattern, i.e., their content is the highest in muscles with a high content of ETC proteins, such as the soleus muscle (mitochondria-rich muscle), and lower in muscles with a lower content of ETC proteins, such as the fast muscles (MGF and TA). Additionally, we found a strong positive correlation between [GSSG] and SOD2 (r = 0.69, *p* < 10^−4^), [GSSG] and ETC content (r = 0.77, *p* < 10^−4^), [GSSG] and myoglobin content (r = 0.62, *p* < 10^−4^).

### 2.5. Correlation between Muscle NO_x_ Concentration and Muscle Protein Content

In the striated muscles, NO_2_^−^ mirrored the distribution of muscle ETC and myoglobin content (Figure 1). Accordingly, a significantly negative correlation (*p* < 0.05) between [NO_2_^−^] and ETC content (Figure 4A), as well as [NO_2_^−^] and myoglobin content (Figure 4B), was found. Moreover, a significantly negative correlation between [NO_2_^−^] and SOD2 content (Figure 4C) was observed in the striated muscles.

Furthermore, a significantly negative correlation between muscle nitrate [NO_3_^−^] and myoglobin content (r = −0.29, *p* = 0.03) was found, whereas no significant correlation (*p* > 0.05) was noticed between [NO_3_^−^] and ETC proteins content, or between [NO_3_^−^] and SOD2 content (not shown).

### 2.6. Correlation between Muscle NO_x_ Concentration and Muscle Oxidative Stress as Well as Antioxidant Capacity

A strong negative correlation was found between [NO_2_^−^] and oxidised glutathione ([GSSG]) (Figure 4D). Moreover, a significantly positive correlation was observed between [NO_2_^−^] and antioxidant capacity of the tissue (Figure 4F), while a clear tendency to positive correlation (*r* = 0.24, *p* = 0.06) was observed between the muscle [NO_2_^−^] and reduced glutathione (Figure 4E). No significant correlation (*p* > 0.05) was recognised between [NO_3_^−^] and [GSSG], [NO_3_^−^] and [GSH], or [NO_3_^−^] and total antioxidant capacity (not shown).

## 3. Discussion

The main finding of the present study was that the nitrite concentration in skeletal muscles differs between the various slow- and fast-twitch muscle types, whereas nitrate concentration did not significantly differ. Specifically, we demonstrated for the first time that [NO_2_^−^] is lower in rat striated muscles–such as heart ventricles and soleus, which possess a higher ETC protein content and a higher myoglobin level (oxidative muscles)–than in the muscles with lower contents of these proteins such as MGF and TA (glycolytic muscles). In addition, we found a strong negative correlation between [NO_2_^−^] and ETC protein contents, as well as negative correlation between [NO_2_^−^] and myoglobin content, supporting the above data. Therefore, our results clearly expand the knowledge of the skeletal muscle NO^•^ pool [10] by showing that the concentration of skeletal muscle nitrite, which is a signalling molecule and a direct substrate for NO^•^ generation under physiological conditions (nitrate–nitrite–NO^•^ pathway), depends on muscle ETC proteins and myoglobin content.

### 3.1. Nitrite and Nitrate Concentration in the Striated Muscles of Varied Muscle Fibre Type Composition

Tissue nitrate and nitrite have been recognised as storage forms of NO^•^ that can be used to maintain NO^•^ bioavailability under challenging conditions (i.e., hypoxia, acidosis and ischemia) when NO^•^ synthesis is impaired in the conventional NOS-dependent pathway [8,13]. Particularly in the heart, the reduction of NO_2_^−^ to NO^•^ has been demonstrated to exert cardioprotective effects in the myocardium, as reflected by reduction of infarct size and improvement of the recovery of ventricular function [31]. The cardioprotective effect of NO_2_^−^ has been found to be limited to the region of insult, which clearly reinforces the need for specific hypoxic and acidotic conditions (related to myocardial ischemia) to generate NO^•^ from NO_2_^−^ [31]. The process of nitrite reduction in the ischemic heart is suggested to involve both protein/enzymatic pathways (with the crucial role of deoxymyoglobin) [32] and non-enzymatic acidic disproportionation [8,33].

The importance of NO_2_^−^ reduction in the locomotory skeletal muscles still needs to be elucidated. However, its beneficial impact on the muscles may be related to the NO^•^-dependent improvement of oxygen delivery during exercise through the mechanism of acidic–metabolic vasodilation [12] and/or hypoxic vasodilation [34]. In addition, it has been postulated that this mechanism might play a role in muscle force production [18]. Taking into account the role of acidosis and hypoxia in the NO_2_^−^ reduction, it could be assumed that this reaction in the skeletal muscles could be physiologically relevant, especially during high-intensity exercise, as it leads to a drop of muscle pH to below 6.4 (from a pH amounting to approximately 7.0 at resting state) under extreme conditions [35].

Interestingly, NO_2_^−^ disproportionation has been shown to accelerate by about 8 times when the pH value decreases from 7.0 to about 6.3 (demonstrated in the reaction catalysed by xanthine oxidoreductase and aldehyde oxidase) [36]. Such a drop in pH could be observed in the skeletal muscles during fatigue [35]. Therefore, the intensification of hydrogen ion accumulation and other factors related to muscle fatigue when performing an exercise in the high-intensity domain, i.e., above the lactate threshold (or, strictly speaking, above the critical power) [29], might facilitate NO_2_^−^ reduction and NO^•^ generation in the skeletal muscles. This might take place especially in the fast-twitch glycolytic muscle fibres characterised by much lower PO_2_ [37], lower pH [38] at basal conditions and greater metabolic disturbances at fatigue than the slow-twitch oxidative muscles [27,28,35].

By investigating various types of striated muscles in the course of this study, such as heart ventricles, the slow-twitch soleus and fast-twitch muscles (MGS, MGF, TA), we aimed to study muscles with different mitochondrial contents and varied capacities of oxidative energy production to relate the level of muscle NO^•^ metabolites ([NO_2_^−^] and [NO_3_^−^]) to the muscle’s oxidative potential (see Methods). As demonstrated in our study, fast-twitch glycolytic locomotory muscles (TA and MGF) exhibited greater NO_2_^−^ content at basal conditions than the slow-twitch oxidative soleus (Figure 1A). Moreover, we have found that the global NOS activity (reflected by the citrulline-to-arginine ratio) was significantly higher in the fast-twitch gastrocnemius (MGS and MGF) than in the slow-twitch soleus (Figure 2A). Surprisingly, we have found significantly lower NOS activity in the TA muscle when compared to MGS and MGF (Figure 2A). This unexpected result might arise from a biochemical and functional differences when compared the TA muscle with the gastrocnemius muscle. Namely, it has been previously reported that NOS protein expression is fibre type-dependent. Specifically, Punkt et al., [39] demonstrated a higher NOS expression in fast oxidative-glycolytic (FOG) fibres than in fast glycolytic (FG) fibres and in slow oxidative (SO) muscle fibres. Furthermore, the content of FOG fibres (with higher NOS expression) has been found to be greater in the gastrocnemius muscle compared to the TA muscle [40]. This can at least partly explain the difference in NOS activity between the gastrocnemius and the TA muscle as observed in our study. Additionally, the obtained results could be affected by some factors related to the pattern of daily muscle activation. Namely, in the freely moving rats the TA (the dorsiflexor) is less activated than the gastrocnemius muscle (the plantar flexor) [41]. This may lead to lower activation of NOS in TA via muscle contraction related factors such as e.g., the changes in intracellular calcium levels. Summing up, the observed in our study lower NOS activity in the TA muscle could be due to lower NOS protein expression as well as to lower muscle activity than in the gastrocnemius muscle, for the reasons as discussed above. Nevertheless, this issue deserves further study.

In addition, we found a clear tendency (*r* = 0.25, *p* = 0.057) to positive correlation between NOS activity and NO_2_^−^ concentration in the striated muscles, which revealed that, under basal conditions (normoxia), tissue NO_2_^−^ originates from NO^•^ produced in the classical NOS-dependent pathway, as suggested previously [13].

Our results–which demonstrated 1.3-fold higher NO_2_^−^ content (relative to mg of protein, detected by high-performance liquid chromatography (HPLC)) in the fast-twitch glycolytic muscles of rats (e.g., MGF, composed of ~77% of type IIB muscle fibres) compared to the slow-twitch oxidative soleus (with no type IIB muscle fibres, see e.g., Staron et al. [40])—are in contrast to the recent data presented by Park et al. [42]. Using chemiluminescence, Park et al. [42] found a negative relation between NO_2_^−^ content and muscle fibre type composition; i.e., higher basal NO_2_^−^ content was found in the soleus when compared to the fast-twitch gluteus (~76% of type IIB muscle fibres). In the case of nitrate, they found no dependence on the muscle fibre type composition [42]. It needs to be underlined that a HPLC-based methodology (used as in our study) is widely used for in vivo studies to determine the NO^•^ metabolites in biological samples [43]. Moreover, our results for the concentrations of NO_2_^−^ and NO_3_^−^ in the heart muscle (~8 pmol and ~400 pmol relative per mg of protein, respectively for [NO_2_^−^] and [NO_3_^−^]), were similar to those obtained by Bryan et al. [9], as both studies used the same HPLC analysis of NO_x_, however, Bryan et al. [9] did not analyse the NO_x_ content in skeletal muscle. In addition, we calculated the NO_2_^−^ and NO_3_^−^ concentrations in muscle homogenates (nmol per g of tissue) and found comparable results to those reported by Park et al. [42] for soleus muscle (0.79 ± 0.29 versus 21.3 ± 14.1 nmol per g of tissue, respectively, for [NO_2_^−^] and [NO_3_^−^]), where, in the case of TA, NO_x_ content was found to be several times (~2–3 fold) higher (0.94 ± 0.24 versus 25.2 ± 13.8 nmol per g of tissue, respectively, for [NO_2_^−^] and [NO_3_^−^]) than in above mentioned study by Park et al. [42]. The apparent discrepancy between the results of Park et al. [42] and our results may be due to differences in the preparation of skeletal muscle tissue for NO_x_ measurement. Namely, when studying a mixed muscle (such as TA, EDL, gastrocnemius) this issue seems to be vitally relevant opposed to more homogenic muscle such as the soleus. Therefore, in our study, special care was taken to clearly dissect the red, slow (MGS) versus white, fast part (MGF) of gastrocnemius muscle, whereas in case of the soleus and TA, the whole muscle was taken and homogenised for NO_x_ analysis.

Our results regarding the differences between the fast-twitch glycolytic and the slow-twitch oxidative muscles in the potential to generate NO^•^ via NO_2_^−^ reduction are in accordance with the notion that both the mechanism of vasodilatation during exercise [25] as well as the impact of NO^•^ on force production [18] differ between skeletal muscles with varied muscle fibre type compositions. Accordingly, it has been reported that selective inhibition of the nNOS (the main NO^•^ synthase in the muscle) by infusion of S-methyl-L-thiocitrulline (SMTC) in rats shortly before high-intensity running significantly reduced muscle blood flow when the rats ran at high speed on a treadmill. This effect was predominantly visible in the fast-twitch glycolytic muscle fibres [44]. In accordance with these results, it has been demonstrated that nitrate supplementation enhances blood flow and muscle oxygen delivery during exercise, preferentially to the parts of muscles composed predominantly of fast muscle fibres [25]. In addition, nitrate supplementation enhances tetanic force production in fast muscle fibres, while not in slow-twitch muscles. This effect might be related to an improvement of calcium handling [18].

In summary, based on our results, one could conclude that, of all locomotory muscles, the fast muscles are more predisposed to generate NO^•^ than the slow-twitch muscles–not only in NOS-dependent pathways, as presented previously [39], but also through the NOS-independent pathway from NO_2_^−^. This is due to the greater content of NO_2_^−^ (as a substrate) at basal conditions as found in the present study, accompanied by specific physiological features of fast-twitch muscle fibres, including lower microvascular partial O_2_ pressure [37], lower pH [38] at basal conditions. In addition, what is even more relevant the fast-twitch muscles develop deeper disturbances in muscle metabolites concentrations, especially deeper decrease in pH at the stage of fatigue compared to slow-twitch muscles [27,28,35], that may contribute to greater generation of NO^•^ from NO_2_^−^ during exercise.

### 3.2. Relationship between the Myoglobin Content and the Nitrite Concentration in Striated Muscles

NO_2_^−^ reduction to NO^•^ occurs via specific tissue NO_2_^−^ reductases, which include proteins involved in oxygen metabolism such as myoglobin engaged in oxygen buffering and transport in the muscle cell [45], and mitochondrial ETC proteins engaged in the oxygen usage and ATP synthesis [17]. It must be emphasised that myoglobin, which is expressed in the striated muscles (heart and skeletal muscles) and in the smooth muscles [34], links oxygen and NO^•^ metabolism as it is involved in both the oxygen buffering process and NO^•^ homeostasis [46,47,48,49]. The role of myoglobin as an oxygen reservoir in the striated muscle has been studied, among others, in the muscles of diving animals undergoing a prolonged apnoea phase. It has been shown that the myoglobin content in their muscles might be 10–30-fold higher compared to terrestrial animals [50]. Moreover, it has been demonstrated that, during exercise, myoglobin becomes desaturated [51], and the desaturation of myoglobin starts immediately with the onset of muscle contraction [52]. Finally, myoglobin may interact with mitochondrial cytochrome-*c* oxidase, suggesting there is direct myoglobin-mediated oxygen transportation to the mitochondria [53].

Despite acting as a storage of oxygen, myoglobin either scavenges NO^•^ (oxidation of NO^•^ to nitrate) or produces NO^•^ (from NO_2_^−^) depending on the oxygen gradient [8,32,49]. Firstly, under normoxic conditions, myoglobin acts as a NO^•^ scavenger (similarly to other ROS), counteracting NO^•^-dependent inhibition of cellular respiration and protecting muscle tissue from energetic imbalance [46]. The protective role of myoglobin as a NO^•^ scavenger in the heart has been demonstrated in the model of myoglobin-deficient mice [47,54]. Deficiency of myoglobin, even accompanied by a transient increase in NO^•^ (i.e., under the condition of iNOS-induced nitrosative stress), resulted in cardiac hypertrophy and development of heart failure [47,50]. On the other hand, under hypoxic and ischemic conditions, myoglobin acts as a NO_2_^−^ reductase and is involved in NO^•^ generation [45,48]. It must be emphasised that the reduction rate of myoglobin-dependent NO_2_^−^ is about 32 times faster than that of haemoglobin [45]. Myoglobin is more efficient as a NO_2_^−^ reductase than isolated mitochondria, as recently shown [55].

The specific distribution of myoglobin in the locomotory muscles (higher content in the slow-twitch type I than in the fast type II muscle fibres) is in accordance with other results published previously [56]. In addition, the higher ETC protein content in the heart compared to the slow-twitch soleus and fast-twitch MGF (~3 and 11-fold, respectively for Sol and MGF) (Figure 1D) is in accordance with data showing that mitochondrial volume in rat hearts amounts to ~28% of the cardiomyocyte’s volume [57], which is about 3-fold greater than in the soleus and even 13-fold greater than in fast glycolytic muscles [58]. Accordingly, in the present study, we reported a significantly positive correlation between myoglobin content and ETC protein content in striated muscle (r = 0.51, *p* < 10^−4^). This correlation underlines the phenomena that the high content of mitochondrial ETC (as in red, oxidative muscles) in normoxia is tightly coupled to high myoglobin content. Moreover, the high myoglobin content in oxidative muscles, which, on the one hand, secures oxygen for maintaining ATP synthesis in OXPHOS (functioning as an oxygen reservoir) and, on the other hand, acts as a NO^•^ scavenger (in normoxia), and protects OXPHOS from NO^•^-dependent inhibition and from the imbalance of energy metabolism [46].

As mentioned above, we found a lower NO_2_^−^ concentration in the red oxidative muscles (Figure 1A), which possess a high myoglobin and ETC protein content (Figure 1C,D). This result clearly demonstrates the importance of OXPHOS for energy metabolism in oxidative muscles. Specifically, under physiological conditions, muscles with high levels of myoglobin and ETC proteins (such as the heart and soleus) and possessing NO_2_^−^ reductase activity [17,45] should be protected from the possibility of excessive NO_2_^−^ reduction to NO^•^, which in turn could inhibit cytochrome-*c* oxidase and thus limit the rate of ATP resynthesis in OXPHOS [46]. Hence, the low substrate availability (nitrite) in the heart and soleus is physiologically justified, taking into account high dependence of energy metabolism in those muscles on ATP resynthesis in OXPHOS (above 90% of energy delivery) [59]. On the other hand, due to fast muscles’ higher capacity of glycolytic ATP fluxes, even partial inhibition of cytochrome-*c* oxidase by NO^•^ generated from nitrite would be less harmful to the muscle cells in terms of energy status than in slow muscles, which are dependent to a much higher degree on the energy generated by OXPHOS. Thus, the strong negative correlation between [NO_2_^−^] and ETC protein content and the negative correlation between [NO_2_^−^] and myoglobin content in the striated muscles at baseline conditions that were found in the present study clearly support the above considerations.

It should be kept in mind, however, that under pathological conditions (myocardial ischemia) the deoxymyoglobin-dependent NO_2_^−^ reduction and NO^•^ generation can protect the heart against its damage through downregulation of myocardial oxygen consumption accompanied by a decrease in cardiac performance (myocardial hibernation) [32] in order to preserve a proper balance between ATP utilisation and ATP supply in the contracting heart [32].

### 3.3. Nitrite Content in Relation to Redox Environment in the Striated Muscles of Varied Muscle Fibre Type Composition

In addition to the muscle-specific determination of NO_x_ content, we wanted to relate the NO_2_^−^ content in muscle to the redox state under basal conditions. Tissue NO_2_^−^ level is regulated in part by NO^•^ bioavailability [13], which in turn is largely governed by the basal redox environment [2,60]. In particular, oxidative and inflammatory stress conditions associated with increased superoxide production effectively decrease NO^•^ bioavailability through the generation of peroxinitrite (from NO^•^ and superoxide), a highly deleterious oxidant [2,60]. Under physiological conditions, neither superoxide nor NO^•^ are particularly toxic because they are efficiently removed by specific scavenger enzymes (e.g., superoxide dismutase, which removes superoxide) and by conversion to nitrate by scavenger proteins such as myoglobin [2,60].

In the present study, we found that oxidative stress in the striated muscles under basal conditions, as reflected by the content of oxidised glutathione ([GSSG]), was the highest in the heart and the lowest in the fast-twitch glycolytic muscles (Figure 3A). Accordingly, the expression of a key antioxidant enzyme that scavenges superoxide, namely the mitochondrial isoform of SOD (SOD2), was found to be the highest in the heart and the lowest in the fast-twitch muscles (Figure 3D). Additionally, we found a strong positive correlation between [GSSG] and ETC content (r = 0.77, *p* < 10^−4^), and [GSSG] and SOD2 (r = 0.69, *p* < 10^−4^). Those correlations clearly support the notion that muscles that work constantly and are composed of mitochondria-rich oxidative muscle fibres (such as the heart and the slow-twitch soleus) generate more ROS [61] but are also predisposed to cope with greater ROS production [61,62]. It should be particularly kept in mind that the heart, which is a continuously activated muscle during daily life and possesses exceptionally high tissue oxygen uptake, also exhibits high ROS production even at basal conditions [63]. In addition, physical capacity increases overall ROS production [61,63] and consequently enhances the antioxidants pool in the heart, including an increase of reduced coenzyme Q, an important antioxidant found in all cell membranes [61].

Interestingly, we found a significantly strong negative correlation between [GSSG] and muscle NO_2_^−^ (Figure 4D), as well as a positive correlation between [NO_2_^−^] and antioxidant capacity which was reflected by the ratio of GSH-to-GSSG (Figure 4F). These results clearly demonstrate that baseline NO_2_^−^ content in the striated muscles depends on the redox state of the tissue, which is similar to NO^•^ bioavailability [2]. Surprisingly, we found a significantly negative correlation between muscle NO_2_^−^ and SOD2 content (Figure 4C). This relationship might indicate that mitochondrial SOD isoform simply reflects the mitochondrial content and, similarly to ETC content, is highest in the heart and oxidative locomotory muscles where NO_2_^−^ concentration is low. A strong positive correlation between SOD2 content and ETC protein content in the striated muscles (r = 0.89, *p* < 10^−4^) clearly supports the above-mentioned consideration.

In conclusion, the results of the present study revealed that skeletal muscles do not share uniform nitrite concentrations when at rest. Specifically, we demonstrate for the first time that fast-twitch glycolytic muscles exhibit higher NO_2_^−^ concentration at rest compared to slow-twitch oxidative muscles, and muscle nitrite concentration is inversely related to their myoglobin and mitochondrial protein content. Thus, our results indicated that fast-twitch muscles have a greater potential to generate NO^•^ via NOS-dependent and NOS-independent pathways than slow-twitch muscles and the heart. This muscle property might be of special importance for fast skeletal muscles during strenuous exercise and/or hypoxia, since it might enhance their muscle blood flow via additional NO^•^ provision (acidic/hypoxic vasodilation) and delay muscle fatigue.

## 4. Materials and Methods

### 4.1. Animals

Twelve adult 4-month-old male Wistar rats (body mass 484 ± 30 g) were involved in the present study. During the experiment, the animals were kept in standard laboratory cages in a room with a 12 h/12 h light/dark cycle, controlled temperature (22 ± 2 °C) and humidity (55 ± 10%). All rats had unrestricted access to standard rat feed (Altromin, 1324) and tap water.

### 4.2. Muscle Tissue Extraction

Heart ventricles (heart) and skeletal muscles (SM) i.e., soleus (Sol), medial gastrocnemius (MG), which was divided by excision into a red part (MGS, slow medial gastrocnemius) and a white part (MGF, fast medial gastrocnemius), and the tibialis anterior (TA) were dissected immediately after animal were killed and frozen in the liquid nitrogen (LN_2_). Heart and SM were used for the assessment of muscle NO^•^ metabolites i.e., [NO_2_^−^], [NO_3_^−^], amino acids concentrations ([L-arginine], [L-citrulline], [L-ornithine]) and oxidation/reduction metabolites such as reduced and oxidized glutathione concentration ([GSH] and [GSSG], respectively). Moreover, the heart and skeletal muscle samples were used to analyse the selected protein expression (see below).

By choosing varied types of muscles for the purpose of this study, such as heart ventricles, as well as the slow-twitch soleus and the fast-twitch skeletal muscles (MGS, MGF, TA), we aimed to collect muscle tissues with different contents of mitochondria and varied capacities of oxidative energy production (see e.g., [57,64]) to relate the level of muscle NO^•^ metabolites i.e., [NO_2_^−^] and [NO_3_^−^] to the muscle oxidative potential. In the group of striated muscles heart possess the highest mitochondrial volume i.e., it accounts for~28% of rat cardiomyocytes volume [57]. In rat skeletal muscles the highest content of mitochondria are found in the fast-twitch oxidative type IIA muscle fibres, then in the slow-twitch oxidative type I muscle fibres and in the fast-twitch oxidative glycolytic type IIX muscle fibres. The lowest content of mitochondria are found in fast-twitch glycolytic IIB muscle fibres (for review see [58]).

In our study the oxidative capacities of the analysed muscles were assessed indirectly by measuring the content of the electron transport chain complex subunits (ETC proteins)—considered as the marker of mitochondrial content [30,65]. Summing up, in the present study we have demonstrated data of measured metabolites in the locomotory muscles from the highest to the lowest ETC proteins content (reflecting indirectly mitochondria content) accordingly in Sol, MGS, TA and MGF. This is in agreement with data presented by Staron et al. [40], who reported the following distribution of the mitochondria-rich, oxidative muscle fibres in rat locomotory muscles (i.e., pure type I and IIA): ~91% in soleus, ~40% in MGS, ~15% in TA and ~0.9% in MGF.

### 4.3. Muscle Sample Preparation for NO^•^ Metabolite and Targeted Metabolomic Analysis

Muscle samples were homogenized in ice-cooled phosphate buffer using Precellys Evolution (Bertin, Montigny-le-Bretonneux, France) or UltraTurax T10 (IKA, Staufen, Germany) homogenizers and were kept on ice during the whole sample preparation procedure. The phosphate buffer was consisted of 50 mM Na_2_HPO_4_ with the addition of butylated hydroxytoluen (BHT, 50 µM), 3-amino-1,2,4-triazole (3-AT, 10 µM) and protease inhibitor cocktail (PIC, 1:100, *v*/*v*) and the pH of the buffer was adjusted to 7.4 using 1 M HCl. After the homogenization process, samples were centrifuged (10,000× *g*, 10 min, 4 °C) and the supernatant was analysed using methods described below to determine the concentration of NO^•^ metabolites and chosen metabolites of main biochemical pathways (GSH, GSSG), amino acids and total proteins. The composition of extraction buffer was developed to reliably quantify all mentioned endogenous molecules and protein concentration as well as to be compatible with applied instruments (LC-MS and ENO-20-NOx systems).

#### 4.3.1. NO^•^ Metabolite Quantification

The levels of NO^•^ metabolites including NO_2_^−^ and NO_3_^−^ in muscle homogenates (Heart, Sol, MGS, MGF, TA) were determined by the application of liquid chromatography-based method with a post column derivatization using an ENO-20-NOx analyser (Eicom Corp., San Diego, CA, USA). Briefly, NO_2_^−^ and NO_3_^−^ were separated on a NO-PAK column (4.6 × 50 mm; Eicom Corp., San Diego, CA, USA) with a subsequent reduction of NO_3_^−^ to NO_2_^−^ using a cadmium–copper column (NO-RED; Eicom Corp., San Diego, CA, USA). In a reaction coil placed in a column oven (35 °C), NO_2_^−^ was mixed with Griess reagent and formed a purple diazo compound were detected at a wavelength of 540 nm. The mobile phase consisted of Carrier Solution and the Griess reagent (Reactors A and B Solution, 1:1, *v*/*v*). Before analysis muscle proteins were precipitated with methanol at the ratio of 1:1 (*v*/*v*), centrifuged at 10,000× *g* for 10 min, and the supernatant injected directly into the system. The level of NO^•^ metabolites measured in muscle samples was normalised to mg of total protein. Nowadays, HPLC-based method combined with post-column derivatization for NO^•^ metabolites determination in biological samples is widely used in in vivo studies [43], and contrary to chemiluminescence method allow for simultaneous quantification of nitrite and nitrate, that is crucial in case of limited amount of material (sample). Additionally, the measured nitrite levels seem to be comparable between these two methods [15].

#### 4.3.2. Targeted Metabolomic Analysis

The concentration of the selected amino acids including [L-arginine], [L-citrulline], [L-ornithine] as well as reduced ([GSH]) and oxidised glutathione ([GSSG]) was determined in muscle samples (Heart, Sol, MGS, MGF, TA) using LC-MS/MS targeted metabolomic method as described previously with minor changes [66]. The concentration of L-arginine and its metabolites L-citrulline, L-ornithine were measured simultaneously in muscle samples. Moreover, the specific ratios of citrulline to arginine and ornithine to arginine were calculated to describe nitric oxide synthase and arginase activities, respectively.

A chromatographic separation of analytes was performed with the aid of UFLC Nexera (Shimadzu, Kyoto, Japan) using Acquity UPLC BEH C18 (1.7 µm 3.0 × 150 mm; Waters, Milford, MA, USA) as an analytical column. The mobile phases consisted of 5 mM HCOONH_4_ (pH 5.0) and ACN:100 mM HCOONH_4_ (95:5 *v*/*v*, pH 5.0) and were delivered in the gradient elution mode.

For the detection QTRAP 5500 mass spectrometer (Sciex, Framingham, MA, USA) with a electrospray interface operated in positive and negative ionisation modes was used. The operating parameters for mass spectrometer used in multiple reaction monitoring method (MRM) were as follows: Curtain Gas: 25 psi, Collision Gas: medium, Temperature: 500 °C, Ion Source Gas 1: 40 arb., Ion Source Gas 2: 50 arb. and Spray Voltage: 5500 V and −4500 V for positive and negative ionisation modes, respectively.

The analytes were extracted from muscle homogenates using 0.5 mL of dry-ice-cold (−70 °C) extraction mixture (acetonitrile:methanol:water 5:2:3, *v*/*v*/*v*) (Witko). The samples were vortexed for 5 min, placed on dry ice for 30 min for protein precipitation, sonicated on ice for 15 min, and centrifuged at 16,600× *g*, 4 °C for 15 min. A supernatant was lyophilized and kept at −80 °C until analysis. The lyophilized samples were reconstituted in 50 µL of water containing internal standards for the metabolites (labetalol, nicotinamide-d_4_ and dextrorphan at the concentration of 1 µg mL^−1^). The samples were injected into LC-MS/MS system twice (5 µL) employing positive and negative ionisation mode. The quantification of studied metabolites was performed using the most intensive and specific ion transitions for analytes and their internal standards. The concentration of studied metabolites measured in muscle samples was normalised to mg of total protein. The specific ratios of citrulline to arginine and ornithine to arginine, was calculated to describe NO^•^ synthase and arginase activities, respectively.

Nitrite and nitrate content as well as amino acids, GSH and GSSG concentrations in muscle homogenates were presented in relation to total protein concentration measured in the same sample. The concentration of total proteins in muscle samples was determined using Pierce™ BCA Protein Assay Kit (Cat#23225, ThermoFisher Scientific, Waltham, MA, USA). All samples were prepared and analysed following the manufacturer instruction. The amino acid standards including L-ornithine (Cat#O2375), L-citrulline (Cat#C7629), L-arginine (Cat#A5006), as well as GSH (Cat#G4251), GSSG (Cat#G4326), labetalol (Cat#L1011), dextrorphan (Cat#UC205), sodium nitrite (Cat#67398), sodium nitrate (Cat#15736) were purchased from Sigma-Aldrich. Standard substance of nicotinamide-d_4_ (Cat#D-3457) was bought from CDN Isotops.

### 4.4. Protein Extraction and Western Immunoblotting Analysis

Heart and skeletal muscles (Sol, MGS, MGF, TA) lysates were prepared using the extraction buffer (62.5 mM Tris pH 6.8, 10% glycerol, 5% SDS), containing protease and phosphatase inhibitor cocktail (Thermo Scientific™, Waltham, MA, USA). Muscle samples were ultrasonicated (UP 50H sonicator; Hielscher Ultrasonics GmbH, Teltow, Germany) on ice. Then samples were incubated for 1 h under gentle agitation (HulaMixer, Invitrogen, Carlsbad, CA, USA) at room temperature and were centrifuged for 30 min at 13,800× *g* at 4 °C. The supernatants were transferred into fresh microcentrifuge tubes. Protein concentration in the samples was measured using a NanoDrop 2000 UV-Vis Spectrophotometer (Thermo Fisher Scientific™, Waltham, MA, USA). Muscle extracts were stored at −80 °C until further analysis.

Heart-derived and SM-derived protein extracts were separated using 4–20% gradient gels (BioRad, Mini-PROTEAN^®^ TGX™ gels, Cat#4561093, Hercules, CA, USA). Equal amounts of total protein were loaded on gels. To eliminate differences between the gels resulting from the unequal transfer, the internal standard i.e., rat muscle sample was applied on each gel (Appendix A). After electrophoresis proteins were transferred onto nitrocellulose membrane (GE™ Healthcare, Amersham Hybond™, Pittsburgh, PA, USA) at a constant voltage (35 V) in transfer buffer at 4 °C. Following the transfer the detection of protein bands on the WB membranes was performed using Ponceau S staining (0.1% *w*/*v* in 5% acetic acid, Cat#P7170-1L, Merck KGaA, Darmstadt, Germany) to ensure equal loading and transfer of proteins (Appendix A). After Ponceau S staining membranes were incubated with the primary antibodies specific to myoglobin (Cat#ab77232, Abcam, Cambridge, UK), subunits of ETC proteins in mitochondria (Cat#ab110413, Abcam, Cambridge, UK) and mitochondrial isoform of superoxide dismutase (SOD2) (Cat#ADI-SOD-111, Enzo, Life Sciences, Farmingdale, NY, USA). After the incubation with primary antibody, membranes were washed and incubated in the secondary antibodies conjugated with horseradish peroxidase. Protein bands were visualized by an enhanced chemiluminescence method (Appendix A and Figure 3A) and data were imaged using GeneGnome 5 Syngene (GenSys 1.2.7.0, Syngene Bio Imaging, Cambridge, UK). Gene Tools Syngene analysis software was used for densitometric analysis. The optical density values obtained for proteins detected in the heart and in the skeletal muscle samples (myoglobin, ETC proteins, SOD2) were normalized to the internal standard and then to the protein content detected at ~40 kDa using Ponceau S staining (Appendix A), as described previously [67]. The results of our WB analysis showing the muscle-type dependent distribution of myoglobin (Appendix A) and mitochondrial proteins (Appendix A) are in accordance with previous results (for review see [58]) showing higher myoglobin and mitochondrial protein content in oxidative muscles (heart, soleus, MGS) when compared to glycolytic muscle (MGF). 

Data of protein content were presented in arbitrary unit (a.u.). For the purpose of this study the electrons transport chain proteins subunits were presented as the sum of the subunit of complex II (CII, Cat#ab14714), complex III (CIII, Cat#ab14745), complex IV (CIV, Cat#ab14705) and complex V (CV, Cat#ab14748) (Appendix A). We have used the sum of the subunits of complexes of ETC (ETC proteins) as a marker of mitochondrial content in the analysed muscle type (Heart, Sol, MGS, MGF, TA). Additionally, we have presented separately the subunit of complex II (Cat#ab14714) and complex V protein (Cat#ab14748) distribution in the analysed striated muscles (Appendix A).

### 4.5. Statistical Analysis

The results obtained in this study are presented as means, standard deviations (SD) and 95% confidence intervals. The data points that deviated from the group means by more than three standard deviations were treated as outliers and excluded from further analysis. Statistical analyses were performed after checking normality of distribution and homogeneity of variance. In case of some variables ([NO_2_^−^], [NO_3_^−^], [GSH], [GSH-to-GSSG], myoglobin and SOD2 content) the original data were transformed to logarithmic scale in order to be able to perform valid analysis of variance. In order to assess the impact of type of striated muscles i.e., heart, soleus, MGS, MGF and TA on the analysed variables one-way ANOVA with post-hoc Tukey test was performed. In case of unequal variances (ETC proteins content, [NO_2_^−^], [GSSG], [GSH-to-GSSG]) Welch ANOVA with post-hoc Games-Howell test was performed to visualise the differences between groups.

Moreover, Spearman’s rank correlation analysis was performed to study the relation between analysed variables in the striated muscles. Statistical analysis was performed using statistical package STATISTICA 13.3 (TIBCO Software Inc., RRID:SCR_014213). Statistical significance was set at *p* = 0.05 and the two-tail *p*-values were presented.

## Figures and Tables

**Figure 1 ijms-23-02686-f001:**
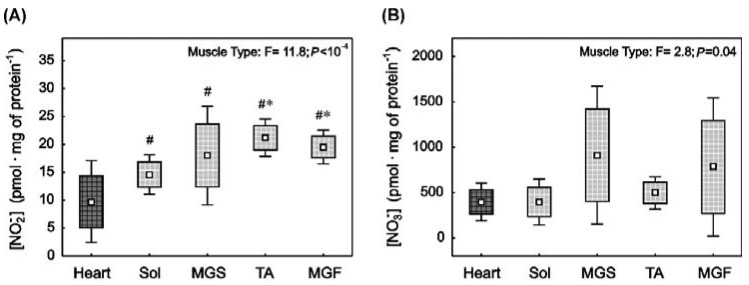
Nitrite, nitrate concentrations and nitrite reductase contents in rat striated muscles with varied muscle fibre type composition. Nitrite ([NO_2_^−^], (**A**)), nitrate ([NO_3_^−^], (**B**)) as well as myoglobin (**C**) and electron transport chain (ETC) proteins content (**D**) in the heart (dark square) and in the locomotory muscles (bright squares) i.e., soleus (Sol), slow part of medial gastrocnemius (MGS), tibialis anterior (TA) and fast part of medial gastrocnemius (MGF). Boxes and whiskers represent, correspondingly, the 95% confidence intervals for means and the standard deviations in samples of 12 animals for each studied muscle tissue (i.e., heart, Sol, MGS, TA and MGF). The impact of muscle type on the analysed variables is presented. Results of one-way ANOVA and Tukey post-hoc are shown in (**C**), whereas Welch ANOVA and post-hoc Games–Howell results are presented in (**A**,**B**,**D**). # denotes a significant difference to the heart; * denotes a significant difference to Sol; ‡ denotes a significant difference to MGS; ◊ denotes significant difference to TA.

**Figure 2 ijms-23-02686-f002:**
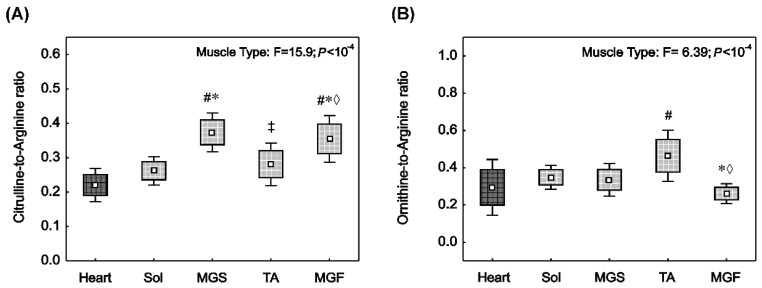
Nitric oxide synthase activity (reflected by citrulline-to-arginine ratio) and arginase activity (reflected by ornithine-to-arginine ratio) in rat striated muscles with varied muscle fibre type composition. Citrulline-to-arginine ratio (**A**) and ornithine-to-arginine ratio (**B**) in the heart (dark square) and in the locomotory muscles (bright squares) i.e., soleus (Sol), slow part of medial gastrocnemius (MGS), tibialis anterior (TA) and fast part of medial gastrocnemius (MGF). Boxes and whiskers represent, correspondingly, the 95% confidence intervals for means and the standard deviations in samples of 12 animals for each studied muscle tissue (i.e., heart, Sol, MGS, TA and MGF). The impact of muscle type on the analysed variables is presented. Results of one-way ANOVA and Tukey post-hoc are shown in (**A**), whereas Welch ANOVA and post-hoc Games–Howell results are shown in (**B**). # denotes significant difference to the heart; * denotes significant difference to the Sol; ‡ denotes significant difference to MGS; ◊ denotes significant difference to TA.

**Figure 3 ijms-23-02686-f003:**
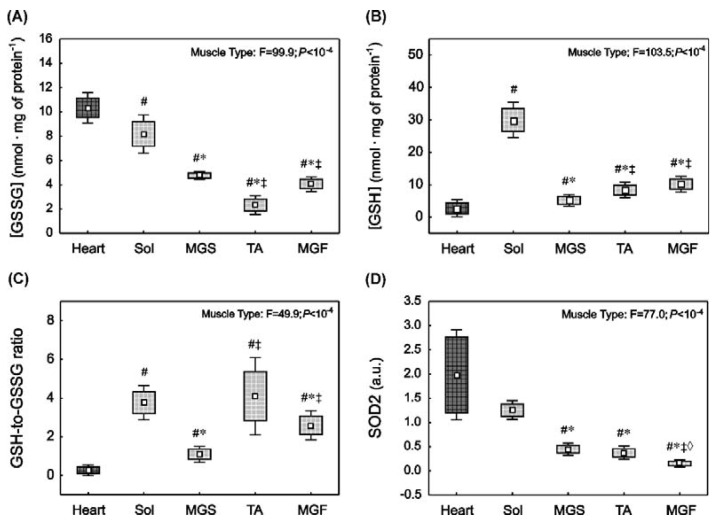
Oxidative stress and antioxidant capacity markers in rat striated muscles with varied muscle fibre type composition. Oxidised glutathione concentration ([GSSG], (**A**)), glutathione concentration ([GSH], (**B**)), total antioxidant capacity ([GSH-to-GSSG], (**C**)) and superoxide dismutase type 2 content (SOD2 content (**D**) in the heart (dark square) and in the locomotory muscles (bright squares) i.e., soleus (Sol), slow part of gastrocnemius (MGS), tibialis anterior (TA) and fast part of gastrocnemius (MGF). Boxes and whiskers represent, correspondingly, the 95% confidence intervals for means and the standard deviations in samples of 12 animals for each studied muscle tissue (i.e., heart, Sol, MGS, TA and MGF). The impact of muscle type on the analysed variables is presented. Results of one-way ANOVA and Tukey post-hoc are shown in (**D**), whereas Welch ANOVA and post-hoc Games–Howell results are presented in (**A**–**C**). # denotes significant difference to the heart; * denotes significant difference to the Sol; ‡ denotes significant difference to MGS; ◊ denotes significant difference to TA.

**Figure 4 ijms-23-02686-f004:**
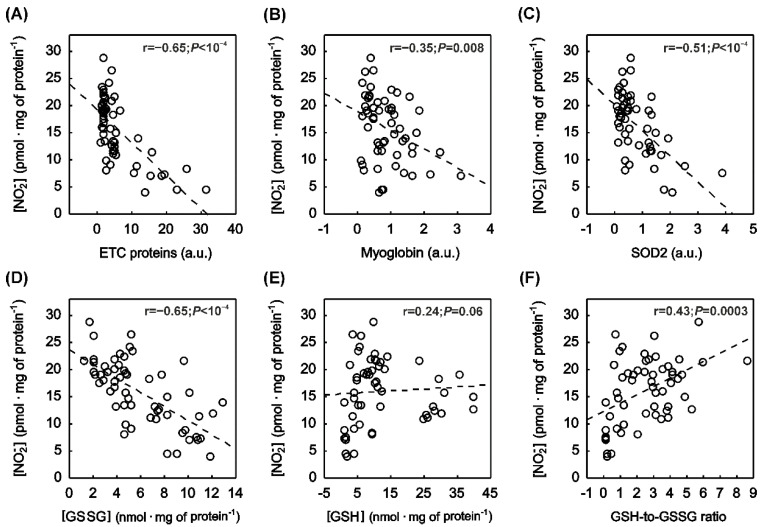
The relationship between muscle nitrite concentration and proteins engaged in oxygen metabolism and redox balance. The relationship between muscle nitrite ([NO_2_^−^]) and the sum of electron transport chain proteins (ETC, (**A**)), [NO_2_^−^] and myoglobin content (**B**), [NO_2_^−^] and superoxide dismutase type 2 content (SOD2, (**C**)), [NO_2_^−^] and oxidised glutathione concentration ([GSSG], (**D**)), [NO_2_^−^] and reduced glutathione concentration ([GSH], (**E**)), and [NO_2_^−^] and antioxidant capacity of the tissue ([GSH-to-GSSG], (**F**)). Dashed lines represent least-squares linear fits based on 60 muscle samples taken from the heart (*n* = 12), Sol (*n* = 12), MGS (*n* = 12), TA (*n* = 12) and MGF (*n* = 12).

## Data Availability

The data supporting reported results are openly available in Jagiellonian University Repository at https://ruj.uj.edu.pl/xmlui/handle/item/287381 [doi:10.26106/fpej-w376] with the access from 24 January 2022–4 February 2024.

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
