# Peer review of "Nitrite Concentration in the Striated Muscles Is Reversely Related to Myoglobin and Mitochondrial Proteins Content in Rats"

_ijms, 2022, doi:10.3390/ijms23052686_

Round 1

Reviewer 1 Report

In this study, the authors evaluated the nitrite/nitrate content in cardiac and skeletal muscle of rats. They assessed both low- and fast-twitch muscles. In addition, they correlated nitrite/nitrate content with the expression level of the myoglobin and mitochondrial electron transport chain proteins.

The aim of this work is interesting.

However, at this stage this study does not provide important novelties to advance in the knowledge of muscle pathophysiology.

Reviewer 2 Report

[NO2]m and [NO3]m what “m” stand for? Please specify. If it stands for muscle, I think it’s unnecessary because you measure NO metabolites only in muscles.

Line 126 “Nitrite concentration (relative to protein concentration)”. Do you mean that you have normalize for total protein concentration?

Lines 130, 163, 173, and other lines in the results section: If you abbreviate soleus with Sol please use only Sol in the Results section.

Line 136 “Our results concerning the content of NO2 − (~8 pmol relative per mg of protein) and NO3 − (~400  pmol relative per mg of protein) in the rat heart were similar to those obtained by Bryan et al. [9].” Please move this sentence to the Discussion section.

Line 140 “In addition, in order to compare our results with the data recently published by Park  et al. [30] concerning NOx content in rat locomotory muscles, we have presented the con-centrations of NO2 − and NO3 − in the Sol and TA homogenates (expressed in nmol per gram of tissue). We found that [NO2 − ]m and [NO3 − ]m in the Sol amounted to 0.79 ± 0.29 versus 143 21.3 ± 14.1 nmol per g of tissue, respectively, whereas in the TA homogenate it amounted to 0.94 ± 0.24 versus 25.2 ± 13.8 nmol per g of tissue, respectively for [NO2 − ]m and [NO3 − ]m.” Please move also this sentence to the discussion section, since there is no figure, supplementary figure or statistical analysis.

Line 189 “However, it was significantly lower compared to fast locomotory muscles: MGS (P = 0.0001), TA (P = 0.07) and MGF (P 190 = 0.0001).” TA has a very peculiar trend, different from other fast muscle fibres, please discuss this result in the discussion section.

Figures: I suggest to revise the colour choice. If there is a trend in colour shades it cannot be appreciated.

How many samples did you use for each measurement? Please specify under each histogram. How many samples did you use for correlation analysis?

Materials and methods: I suggest to move section 4.4. “Protein extraction and Western immunoblotting analysis” in supplementary data since no results of WB are shown in the main text.

Round 2

Reviewer 1 Report

The revised version of the manuscript has been sufficiently improved.

These results can support further studies in the context of muscle pathophysiology.